

# *Dichocarpum hagiangense*—a new species and an updated checklist of Ranunculaceae in Vietnam

Minh Ty Nguyen[1,*], Ngoc Bon Trinh[2], Thanh Thi Viet Tran[3], Tran Duc Thanh[4], Long Ke Phan[3] and Van The Pham[5,6,*]

[1] Institute of Applied Technology, Thu Dau Mot University, Thu Dau Mot, Vietnam
[2] Department of Silviculture foundation and Forest Phytodiversity, Silviculture Research Institute, Vietnamese Academy of Forest Sciences, Ha Noi City, Vietnam
[3] Vietnam National Museum of Nature, Vietnam Academy of Science and Technology, Ha Noi, Vietnam
[4] Southern Center of Application for Forest Technology & Science, Forest Science Institute of South Vietnam, Thu Dau Mot, Vietnam
[5] Environmental Engineering and Management Research Group, Ton Duc Thang University, Ho Chi Minh, Vietnam
[6] Faculty of Environment and Labour Safety, Ton Duc Thang University, Ho Chi Minh, Vietnam
[*] These authors contributed equally to this work.

## ABSTRACT

*Dichocarpum hagiangense* from Ha Giang province, northern Vietnam is described and illustrated. Diagnostic features of the new species are a short rhizomatous stem, (2–)3-foliolate or simple leaves, and pink-purple flowers. The described species is distinct from closely allied *D. trifoliolatum* in having longer sepals, shape and obcordate apex of petal limbs, shorter flower stem, number and tooth shape of basal leaves; it differs from *D. basilare* and *D. carinatum* in having stem leaf, retuse apex and longer of central leaflet, number and (2–)3-foliated (or simple) of leaf. With the support of molecular data, the new species was clearly distinguished from other species in the *Dichocarpum* group by eight autapomorphic characters in nrITS sequence. A key to all species of *Dichocarpum* is provided. We suggest the IUCN conservation status of *D. hagiangense* to be "Critically Endangered". A newest checklist of the family Ranunculaceae in Vietnam is updated.

Corresponding author
Van The Pham, phamvan-the@tdtu.edu.vn

## INTRODUCTION

The flowering plant family Ranunculaceae comprises about 60 genera and 2,500 species worldwide distribution but mainly in East Asia (*Tamura, 1993*; *Wang et al., 2001*). In Vietnam, Ranunculaceae has the presence of 11 genera and about 40 species (*Finet & Gagnepain, 1907*; *Gagnepain, 1938*; *Pham, 1999*; *Nguyen, 2003*).

The genus *Dichocarpum* W. T. Wang et Hsiao (1964: 323) (Ranunculaceae) includes ca. 19 species widely distributed across eastern Asia ranges from the eastern Himalayas to Japan (*Hsiao & Wang, 1964*; *Tamura & Lauener, 1968*; *Fu, 1988*; *Tao, 1989*; *Tamura, 1993*; *Tamura, 1995*; *Fu & Robinson, 2001*). Recently, plus two new species, *D. lobatipetalum Wang & Liu (2015)*: 275) and *D. wuchuanense* S.Z. He (2015: 71) were described from

China, the total species of the genus were increased (*Jiang et al., 2015*; *Wang & Liu, 2015*). However, a little while later, two names, *D. lobatipetalum* and *D. malipoense* were both combined with *D. hypoglaucum* Wang & Hsiao (1964: 327). At a recent time, based on four DNA regions, *Jiang et al. (2015)*, and *Xie, Yuan & Yang (2017)* excluded 18 species including *D. lobatipetalum* and *D. hypoglaucum*. A phylogenetic analysis of the remaining species and taxonomic revision with morphological descriptions of the three complex species (*D. lobatipetalum*, *D. malipoense*, and *D. hypoglaucum*) are needed to improve in furture. Within 19 species, nine species appear in mainland China, one is found in Taiwan, one is recorded in eastern Himalayas, and eight occur in Japan (*Tamura, 1995*; *Fu & Robinson, 2001*; *Jiang et al., 2015*; *Wang & Liu, 2015*; *Xiang et al., 2017*).

In Vietnam, some specimens of Ranunculaceae with the same label (No. 3725) have been collected by P.A. Pételot since 1930 from Sa Pa town, Lào Cai province and deposited in Muséum national d'Histoire naturelle [MNHN-P-P00194832, MNHN-P-P00194833]. The specimens were first identified as *Isopyrum adiantifolium* Hook.f. & Thomson (1855: 42) (*Gagnepain, 1938*), but were later determined as *I. sutchuenense* Franch (1894: 284). In 1973, *Lauener* defined these specimens as *Dichocarpum sutchuenense* (Franch.) W.T. Wang & P.K. Hsiao (1964: 328). In "Cay co Viet Nam: an illustrated of flora of Vietnam", *Pham (1999)* only recorded this species (Fig. 1). After a botanical exploration in Ha Giang province in 2001, *Phan, Averyanov & Nguyen (2001)* discovered *D. dalzielii* (J.R. Drumm. & Hutch.) W.T. Wang & P.K. Hsiao in a cloud forest at an elevation of about 1,500 m a.s.l. (Fig. 1). In addition, in 2002, Averyanov, Loc, and Doan found an unknow *Dichocarpum* species in Van Ban district, Lao Cai province (Fig. 1). The plants had light blue-violet flowers growing on open wet granite rocks of a high waterfall at elevation 1300 m a.s.l. The specimens depositing at HN, LE, and MO should be examined (*HAL 2212*). To date, there are only two species of *Dichocarpum* recorded in Vietnam (*Pham, 1999*; *Phan, Averyanov & Nguyen, 2001*). The genus is still scarcely known in the country.

During fieldwork in the Ha Giang province in northern Vietnam, in the same region of distribution of *D. dalzielii*, a small population of an unknown Ranunculaceae species was discovered. The specimens had a short rhizomatous, unbranched stem, simple or (2–)3-foliolate leaves, two to six flowered inflorescence, five golden-yellow petals and much smaller than sepals, and carpels connate at the base. These characteristics suggested that the specimen was a member of *Dichocarpum*. Detailed studies revealed that some characteristics of the newly collected species did not fit any of the previously reported *Dichocarpum* species described from Vietnam (*Pham, 1999*; *Phan, Averyanov & Nguyen, 2001*), China, or Japan (*Hsiao & Wang, 1964*; *Tamura & Lauener, 1968*; *Fu, 1988*; *Tao, 1989*; *Tamura, 1993*; *Tamura, 1995*; *Fu & Robinson, 2001*; *Jiang et al., 2015*; *Wang & Liu, 2015*; *Xiang et al., 2017*; *Xie, Yuan & Yang, 2017*). Furthermore, it showed substantial morphological differences from closely allied species, *D. trifoliolatum* W.T. Wang & P.K. Hsiao (1964: 324), *D. basilare* W.T. Wang & P.K. Hsiao (1964: 325), and D. carinatum D.Z.Fu (1988: 258) reported from China. Thus, we describe and illustrate this plant as a new species.

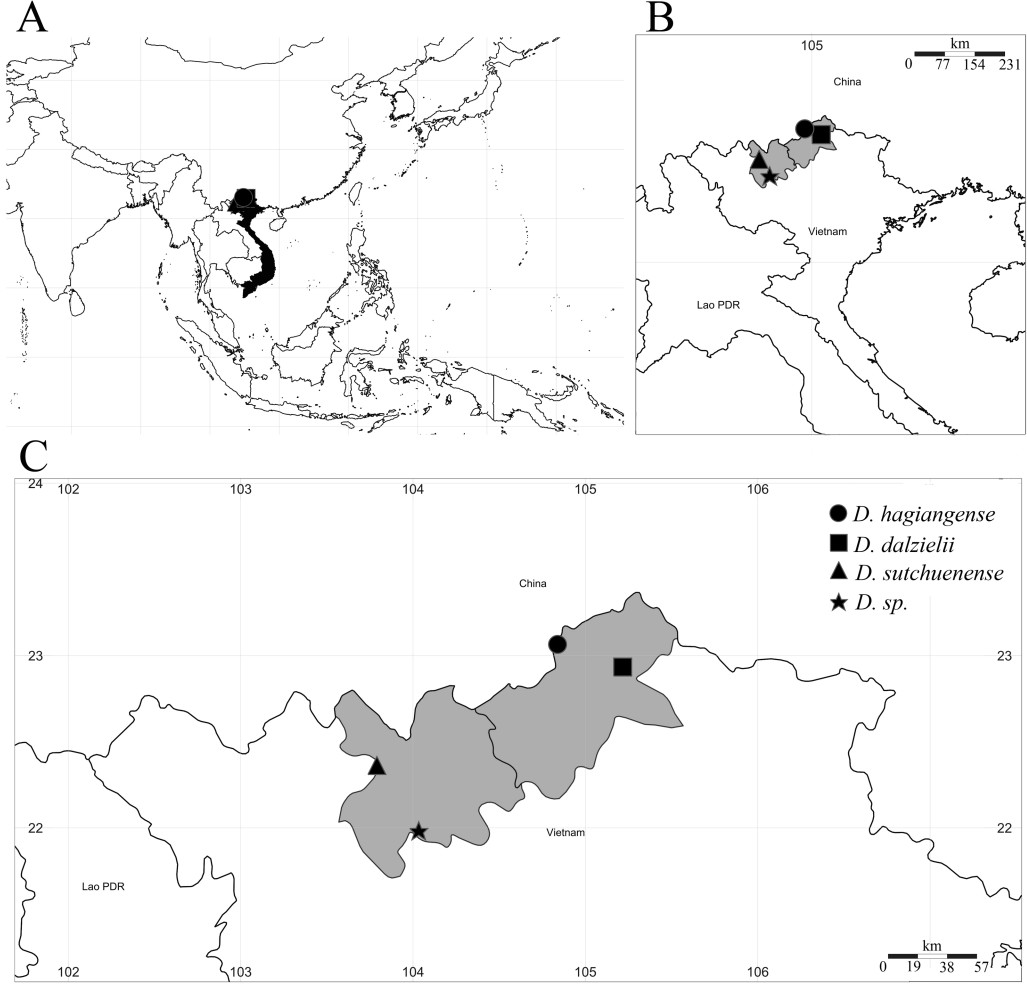

**Figure 1** **Distribution map of *Dichocarpum* species in Vietnam.** (A) Southeast Asia, Vietnam and collection point in black; (B) Northern Vietnam, Ha Giang and Lao Cai provinces in grey; (C) Detail of Ha Giang and Lao Cai provinces, the collection and recorded points in black.

## MATERIALS & METHODS

### Sample collection and morphological analysis

The *Dichocarpum* specimens were collected in natural habitat in March 2018 and June 2020. Collection and fixing specimen procedures were followed the usual procedures for botanical specimens (*Liesner, 1995*; *Maden, 2004*). Morphological descriptions follow *Hsiao & Wang (1964)*, *Radford et al. (1974)*, *Fu (1988)*, *Tamura (1993)*, *Tamura (1995)*, *Fu & Robinson (2001)*, *Harris & Harris (2006)*. The study was based on literature *Hsiao & Wang (1964)*, *Tamura & Lauener (1968)*, *Fu (1988)*, *Tao (1989)*, *Tamura (1995)*, *Fu & Robinson (2001)*, *Pham (1999)*, *Phan, Averyanov & Nguyen (2001)*, *Jiang et al. (2015)*, *Wang & Liu (2015)*, *Xiang et al. (2017)*, *Xie, Yuan & Yang (2017)* and the analysis of specimens at HN, HNU, VNM, VAFS, and LE, MO, P virtual herbaria (acronyms according to *Thiers (2015)*. The distribution map of *Dichocarpum* species in Vietnam was made with SimpleMappr based

on literature of *Phan, Averyanov & Nguyen (2001)* and voucher specimens of LE, MO, P. Conservation analysis was performed using criteria from the International Union for the Conservation of Nature (*IUCN Standards and Petitions Subcommittee, 2019*). The Extent of Occurrence (EOO) and Area of Occupancy (AOO) of each species were estimated using GeoCat (*Bachman et al., 2011*). Collection permits were issued by the "Forest Protection Department of Ha Giang province" (applied by Fauna & Flora International - Vietnam Programme, no. 12/CV-FFI).

## DNA extraction and sequencing

Total DNA was extracted from dried leaves using the DNeasy Plant Minikit. The ITS region was amplified using the forward primer dichFb 5′-CCT GCT CAA GCA GAA CGA C-3′and dichRb 5′-TTG ACA TGC TTA AAT TCA GC-3′designed based on the ITS sequence of *Dichocarpum* spp. obtained from GenBank. The PCR protocol comprised an initial denaturation at 95 °C for 3min, 35 cycles of 50s at 95 °C, 40 s annealing temperature for the primer at 51 °C, 50s extension at 72 °C, and 10min final extension at 72 °C, then 4 °C until used. After purification, DNA fragments were sequenced with a BigDye Terminator Cycle Sequencing Ready Reaction kit and run on an ABI PRISM 3100 Genetic Analyzer. The sequence was deposited in Genbank under accession number MT739412. The ITS sequence of *D. hangiangensis* was aligned using Clustal X 1.64 (*Thompson et al., 1997*) with ITS sequences of other species of *Dichocarpum* and *Isopyrum manshuricum* (EF437119) used as outgroup taxa (*Xiang et al., 2017*). The distance and equally weighted maximum parsimony (MP) and maximum likelihood (ML) analyses were performed using PAUP* (4.0 beta ver.) (*Swofford, 1998*). A heuristic search procedure was used with the following settings: ten replicates of random taxon addition, tree-bisection reconnection branch swapping, multiple trees retained, no steepest descent, and accelerated transformation. Gaps were treated as missing data, and there were no indels within the alignment for the *Dichocarpum* spp. sampled. Bootstrap analysis was carried out with 100 replicates. For ML analysis, the substitution model that best fitted the data set was determined by the Akaike information criterion (AIC) with MODEL Test 3.7 (*Posada & Crandall, 1998*). Bootstrap analysis with 100 replicates was conducted to assess the degree of support for ML tree clades.

## Checklist preparation

The updated checklist is prepared by reviewing all scientific names of Ranunculaceae which had recorded in Vietnam from mainly four monographs –"Flore générale de l'Indo-Chine 1" (*Finet & Gagnepain, 1907*), "Supplément a la flore générale de l'Indo-Chine 1" (*Gagnepain, 1938*), "Cây có Việt Nam: an Illustrated Flora of Vietnam 1" (*Pham, 1999*), and "Checklist of Plant Species of Vietnam 2" (*Nguyen, 2003*). The most widely accepted classification system, APG4 (*Chase et al., 2016*) is applied for the checklist. All the scientific names were nomenclature checked according to Shenzen code of International Association for Plant Taxonomy (*Turland et al., 2018*) together with online consulted from World Flora Online, The Plant List, and International Plant Names Index websites. The invalid names and cultivation species are not recorded in the checklist.

# RESULTS

## Molecular characteristics

The length of the *Dichocarpum* + outgroup taxa ITS sequence alignment was 608 base pairs. MP analysis of this alignment indicated that among 608 characters, 101 were parsimony informative. The phylogenetic trees obtained from MP (tree length 240) and ML (DNA model = GTR+G model, Ln likelihood = −2191.48215), had similar topology (Fig. 2). In the phylogenetic tree, *D. hagiangense* was clustered with *Dichocarpum* group (Fig. 2) including *D. arisanense*, *D. franchetii*, *D. adiantifolium*, *D. basilare*, *D. trifoliolatum*, *D. carinatum*, *D. sutchuense*, *D. auriculatum*, *D. dalzielii* and *Dichocarpum* sp. (*Xiang et al., 2017*).

The pairwise divergence between *Dichocarpum hagiangense* and *Dichocarpum* group ranged from 0.3 to 5.9% (Table 1). *Dichocarpum hagiangense* was clearly distinguished from other species in the group by 8 autapomorphic characters (Supplemental Information).

Key to species of *Dichocarpum*

1. Basal leaves present……………………………………………………...…………...…......… 2
2. Sepal white with purple striation; petal limb bilobed reflexed …………….. *D. dicarpon*
2ı. Sepal white, yellow of pink; petal not reflexed.………....…...……………….……..…3
3. Basal leaves simple, or 1–3-foliolate …………………….……………….…..…4
4. Leaflet margin 3-loped ……………………………………………….……………………5
5. Terminal leaflet 15-20 mm long; seed smooth …………………………....*D. hakonense*
5ı. Terminal leaflet 5-15 mm long; seed granular-roughened or dorsally slightly ridged …………………………………………………………………..6

6. Sepal elliptic; seed 1 mm in diam., granular-roughened ……………..*D. trachyspermum*
6ı. Sepal narrowly ovate; seed ca. 0.75 mm in diam., dorsally slightly ridged …………………………………………………..……*D. arisanense*

4ı. Leaflet margin crenate or coarse teeth ………………………..………………….…..… 7
7. Central leaflet $6-14 \times 3-6.5$ cm, apex attenuate; sepal white …………. *D. wuchuanense*

7ı.	Central leaflet 3.0 −4. 0 × 2.4−2.8 cm, apex retuse; sepal pink purple, pinkish...........8

8.	Inflorescence dichasial; flower diam. 2.0 −2.3 cm; petal limp broadly obcordate ………………………………………………………………….. *D. hagiangense*

8ı.	Inflorescence monochasial; flower diam. ca. 0.7 cm; petal limp flabellate ……………………………………………………….………. *D. trifoliolatum*

3ı.	Basal leaves 5–15-foliolate (rarely 3-foliolate in *D. basilare*)…………….…………9

9.	Leaflet apex long acuminate …………………………………….……..… *D. hypoglaucum*

9ı.	Leaflet apex obtuse, rounded or retuse ………....………………………..….………10

10.	Leaflet apex retuse ………………….……………………………….………. 11

11.	Leaflet suborbicular to subflabellate, apically 5-toothed; flower diam. 4.2–6 mm, stamens 20–45 ……………………………………………..…….…… *D. franchetii*

11ı.	Leaflet broadly rhomboid, apically slightly lobed; flower diam. 6–10 mm, stamens 5–10 ………………………………………………………………… *D. adiantifolium*

10ı.	Leaflet apex obtuse, rounded ……………………………….…………………....12

12.	Leaflet margin 3–5-lobulate or toothed …………...…………………...…………13

13.	Central leaflet subrhombic to rhombic-ovate …………...………………..…*D. carinatum*

13ı.	Central leaflet reniform to flabellate, suborbicular-obovate to flabellate-obovate ………………………………………………………….……...…14

14.	Petal limb funnelform; stamens 10 ………….………………………………….. *D. fargesii*

14ı.	Petal limb suborbicular; stamens 20-45 ……..………….………...……….. *D. sutchuenense*

12ı.	Leaflet margin distally crenate or lobulate ……………….…....…....………….……...15

15.	Basal leaves 11–15-foliolate …………….……..……………………..…………*D. dalzielii*

15ı.	Basal leaves 5-foliolate, rarely 3-foliolate ………………………..…………..……...16

16.	Stem leaves present; follicles 11–15 mm long ……………….…..………………17

17.	Sepal yellow, oblonga obtusa; petal limb peltate-saccate………………………….*D. pterigionocaudatum*

17ı	Sepal white, obovate-elliptic; petal limb broadly obovate …………….. *D. auriculatum*

16ı.	Stem leaves absent; follicles 7.5–10 mm long ………………….………. *D. basilare*

1ı.	Basal leaves absent …………………………………………………………....18

18.	Terminal leaflets broadly ovate, retuse; sepal yellowish-white……….*D. numajirianum*

18ı.	Terminal leaflets cuneate-obovate or cuneate-oblong, obtuse to sub rounded, or rhomboid-ovate; sepal creamy yellow, pale greenish-yellow, or sometime with a purple hue …………………………………………..…………………….…...19

19.	Inflorescences 2–3-flowered; flowers diam. 12-15 mm; petal limb entire, reflexed ………………………………………………………… *D. stoloniferum*

19ı.	Inflorescences few-more than 10-flowered; flowers diam. 7-10 mm; petal limb bilabiate, not reflexed ……………………………………………..….. *D. nipponicum*

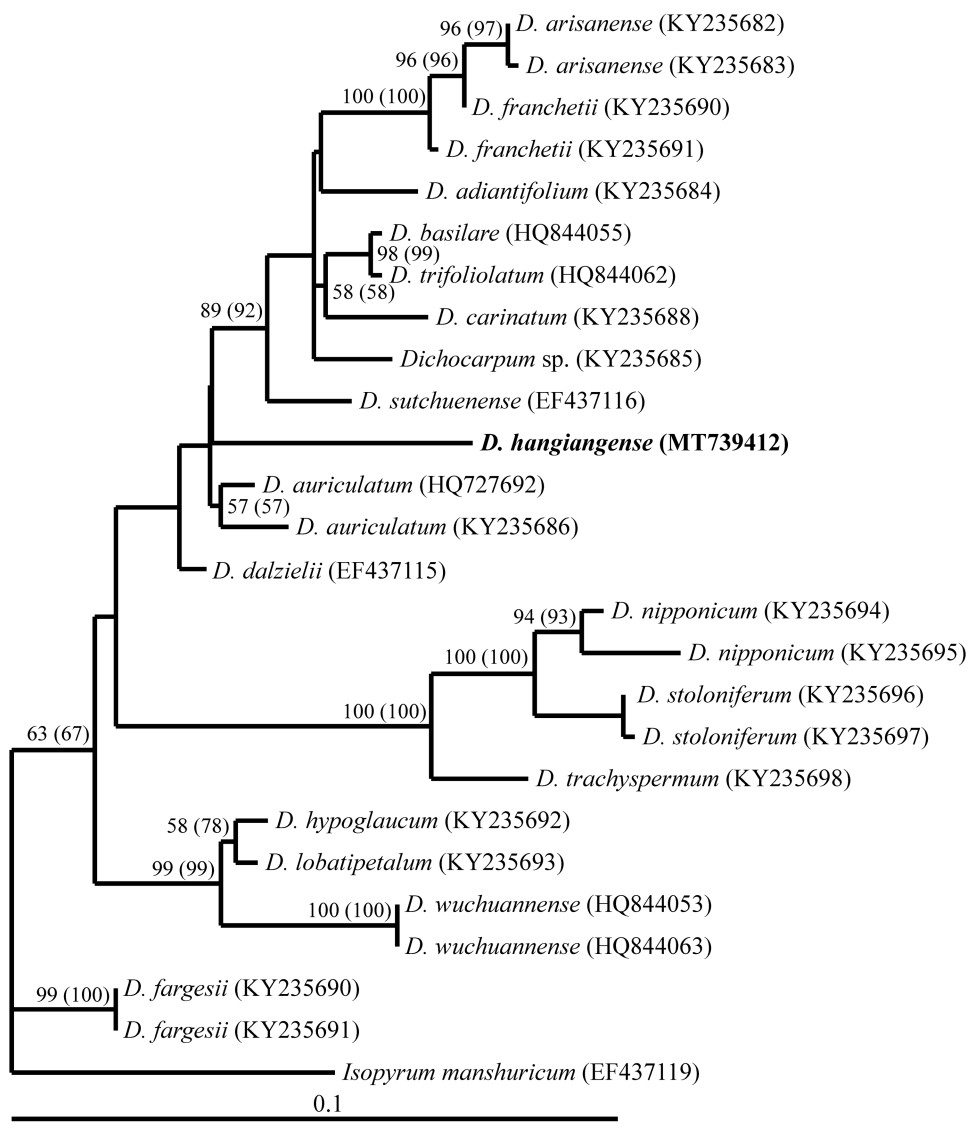

**Figure 2** **A single maximum likelihood tree based on ITS sequences (Ln likelihood = −2191.48215, GTR+G model of DNA evolution) obtained from analysis of the alignment of *Dichocarpum hagiangense* with other sequences of *Dichocarpum* spp. (X).** Bootstrap values are given in appropriate clades. Bootstraps for MP are in brackets. Scales indicate the number of nucleotide changes.

## Species description

*Dichocarpum hagiangense* **L.K Phan & V.T. Pham, sp. nov.**

(Figs. 3, 4 and 5)

### *Type*

Vietnam. Ha Giang: Tung Vai commune, Quan Ba district, forest on limestone mountain, 1,297 m, 23°03′53′N 104°50′20′E, 19 March 2018, *Pham Van The* and *Trinh Ngoc Bon, TB060* (Holotype: VNM-VNM00023655; Isotype: HNU!) (Fig. 5).

*Paratype*

Vietnam. Ha Giang: Tung Vai commune, Quan Ba district, primary evergreen broad-leaved very humid forest, 1,200–1,400 m, around point 23°03′42′N 104°50′42′E, 22 April 2018, *Averyanov et al., VR607* [LE-LE01049587].

**Diagnosis**. According molecular characters new species belong to sect. *Dichocarpum*, subsect. *Dichocarpum. Dichocarpum hagiangense* is morphologically most similar to *D. trifoliolatum*, but differs in having longer sepals, shape and obcordate apex of petal limbs, shorter flower stem, number and tooth shape of basal leaves. However, *D. hagiangense* differs from *D. basilare* and *D. carinatum* in having stem leaf, retuse apex and longer of central leaflet, number and (2–)3-foliated (or simple) of leaf.

**Description.** Perennial herb, glabrous. Rhizome stout, creeping and ascending, 4–9 cm, 0.5–0.8 cm in diam., densely scaly, unbranched; scales green-black when fresh, gray-black when dry, broadly ovate, 2–3 × 5–6 mm, apically rounded. Basal leaves 4–6, (2–)3-foliolate or sometimes simple, slightly thick, abaxially whitish green, adaxially dark green, apically toothed; abaxial veins inconspicuous, adaxial veins distinct; petiole cylindrical, 3.3–10.5 cm, 1–1.5 mm in diam.; 3-foliolate compound leaves with leaflet base cuneate, margin distally crenate, apex retuse, lateral leaflets obliquely rhombic 2.5–3.6 × 0.8–2.4 cm, petiolule 0.3–0.7 cm long, ca. 1.2 mm in diam., grooved, central leaflet rhombic-ovate 3.0–4.0 × 2.4–2.8 cm, petiolule 0.7–1.8 cm long, ca. 1.2 mm in diam., grooved; 2-foliolate compound leaves with leaflets unequal in size, lower leaflet obliquely rhombic, 3.6–5 × 1.8–3 cm, base cuneate to broadly cuneate, petiolule 0.3–0.7 cm long, 1.2–1.5 mm in diam., grooved, upper leaflet obliquely rhombic or semi-orbicular, 4–7 × 2.5–5.8 cm, base cuneate or oblique, petiolule 0.6–0.8 cm long, 1.2–2.0 mm in diam., grooved; simple leaves with leaf blade nearly orbicular, broadly ovate or broadly cuneate, 2.9–5.2 × 2.5–5.4 cm, base rounded. Stem leaves 2–3, 3-lobed or entire, smaller than basal leaves, petiole ca. 1–2 mm long and 0.5 mm in diam., winged, central leaflet ca. 2× 2 cm, lateral leaflets and simple leave ca. 1× one cm. Flowering stem cylindrical, 9.5–14.5 cm tall, ca. 1.5 mm in diam. Inflorescences dichasial, 2–6-flowered; bracts foliaceous, opposite, rounded, ca. 0.4 × 0.4 cm, petiole ca. 0.5 mm. Flowers 2.0–2.3 cm in diam., glabrous; pedicel 1.7–9.0 cm; sepals 5, pink-purple, elliptic to oval, 10.5–11.5 × 5.5–7.0 mm, apex obtuse; petals 5, petal limbs broadly obcordate, golden-yellow, apex obcordate, 1.2–1.4 × 1.6–1.8 mm, claw 1.8–2.3 mm long; stamens ca. 30–40, 3 –4 mm; anthers broadly ellipsoid, ca. 0.8 × 0.6 mm. Ovary 2–3-carpels, free, base connate, narrowly oblong, ca. 5.5 × 1 mm; follicles 2–3, narrowly oblong, sessile, 10–14 mm long; persistent styles ca. two mm long. Seeds 14 or 15 (sometime up to nine regenerate seeds), yellowish dark green, globose, ca. 0.7 mm in diam., smooth. Flowering and fruiting in March to April.

**Phenology.** Flowering and fruiting were observed in March to April.

**Habitat and ecology.** The new species grows in disturbed primary evergreen forest on a limestone mountain at elevations of 1297 m, as a lithophytic herb on large wet mossy boulders and cliffs on steep slopes (Fig. 4).

**Distribution and Conservation status.** *Dichocarpum hagiangense* was only recorded from one small population in Ha Giang province of Vietnam (Fig. 1). The existing population is facing the risk of extinction in the wild, since the area where this species is found does

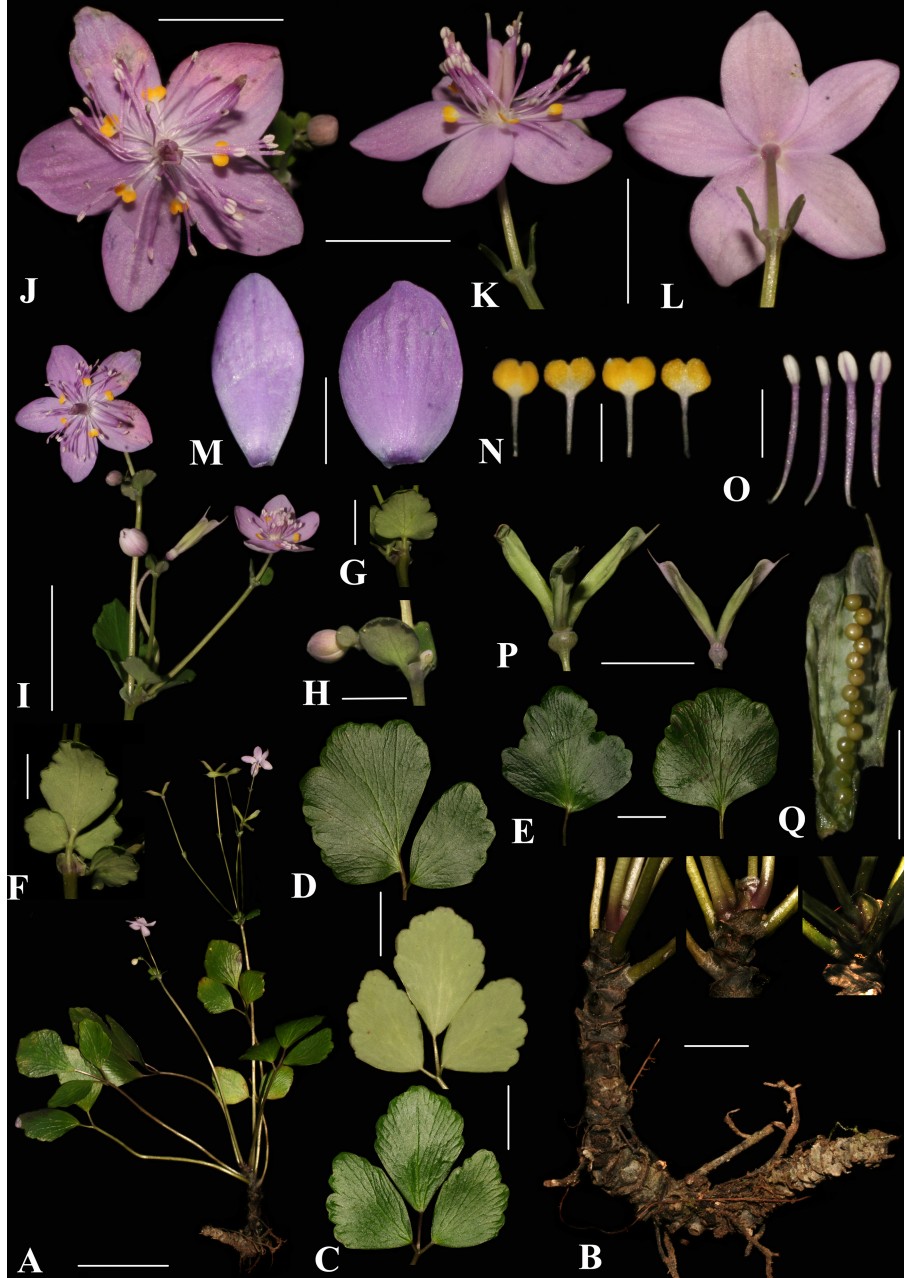

**Figure 3** *Dichocarpum hagiangense* L.K. Phan & V.T. Pham. (A) Flattened flowering specimen. (B) Scaly rhizomes. (C) 3-foliolate leaves adaxial (lower) and abaxial (upper) views. (D) 2-foliolate leaf adaxial view. (E) Single leaves abaxial view. (F, G) Stem leaves. (B) Bract. (I) Inflorescences. (J) Flower frontal view. (K) Flower side view. (L) Flower behind view. (M) Sepals frontal view. (N) Petals. (O) Stamens. (P) Follicles. (Q) Seeds. Scale bars: A = 5 cm; B, C, D, E = 2 cm; F, G, H, M, Q = 0, 5 cm; I, J, K, L, P = 1 cm; N, O = 0, 2 cm.

not belong to any protected forest. The habitat is highly disturbed by the local people for cardamom and *Lysimachia foenum-graecum* cultivations, collecting timber, firewood and non-timber forest products. The species is very rare and only known from one population

**Table 1** Pairwise distance between taxa in *Dichocarpum hagiangense* and closely *Dichocarpum* species (below diagonal: total character differences, above diagonal: mean character differences adjusted for missing data).

| | Species | 1 | 2 | 3 | 4 | 5 | 6 | 7 | 8 | 9 | 10 | 11 |
|---|---|---|---|---|---|---|---|---|---|---|---|---|
| 1 | *D. hagiangense* | – | 4,9 | 5,4 | 5,3 | 5,9 | 5,8 | 5,7 | 5,2 | 4,4 | 5,2 | 5,4 |
| 2 | *D. dalzielii* | 29 | – | 3,7 | 3,7 | 4,1 | 4,1 | 5,2 | 4,6 | 1,5 | 2,7 | 3,4 |
| 3 | *D. basilare* | 32 | 22 | – | 0,3 | 2,5 | 2,4 | 3,7 | 3,1 | 3,2 | 2,9 | 2,2 |
| 4 | *D. trifoliolatum* | 31 | 22 | 2 | – | 2,5 | 2,4 | 3,7 | 3,1 | 3,2 | 2,9 | 2,2 |
| 5 | *D. adiantifolium* | 35 | 24 | 15 | 15 | – | 2,5 | 4,1 | 3,4 | 3,6 | 3,2 | 2,7 |
| 6 | *D. carinatum* | 34 | 24 | 14 | 14 | 15 | – | 4,4 | 3,7 | 3,6 | 3,4 | 2,5 |
| 7 | *D. arisanense* | 34 | 31 | 22 | 22 | 24 | 26 | – | 0,7 | 4,7 | 4,7 | 4,1 |
| 8 | *D. franchetii* | 31 | 27 | 18 | 18 | 20 | 22 | 4 | – | 4,1 | 4,1 | 3,4 |
| 9 | *D. auriculatum* | 26 | 9 | 19 | 19 | 21 | 21 | 28 | 24 | – | 2,2 | 3,1 |
| 10 | *D. sutchuenense* | 31 | 16 | 17 | 17 | 19 | 20 | 28 | 24 | 13 | – | 3,1 |
| 11 | *Dichocarpum* sp. | 32 | 20 | 13 | 13 | 16 | 15 | 24 | 20 | 18 | 18 | – |

of less than 50 mature individuals, in a habitat that is seriously threatened. According to *IUCN Standards and Petitions Subcommittee (2019)* criteria B1ab(ii) + B2ab(ii), with EOO (Extent of Occurrence) = 0 km$^2$ and AOO (Area of Occupancy) = 4.000 km$^2$, this species should be classified as "critically endangered" (CR).

**Etymology.** The species epithet 'hagiangense' refers to Ha Giang province, the only site where the species is currently known.

## Remarks

Based on our molecular data and according to the phylogenetic study of *Xiang et al. (2017)*, the described species belongs to *Dichocarpum* subclade of *Dichocarpum* section in clade II which clade includes two sections including *Fargesia*. The new species is closely allied to *D. trifoliolatum*, *D. basilare,* and *D. carinatum* from China in habit and some morphologic characteristics, such as the present rhizome, few leaves, central leaflet shape, or sometimes 3-flowered inflorescence. However, *D. hagiangense* differs from *D. trifoliolatum* in having longer sepals, shape and obcordate apex of petal limbs, shorter flower stem, number and tooth shape of basal leaves. However, *D. hagiangense* differs from *D. basilare* and *D. carinatum* in having stem leaf, retuse apex and longer of central leaflet, number and (2 −)3-foliated (or simple) of leaf. A detailed comparison between *D. hagiangense* and related species, *D. trifoliolatum*, *D. basilare*, and *D. carinatum* are given in Table 2.

**Additional material examined**. Vietnam. Ha Giang: Tung Vai commune, Quan Ba district, forest on limestone mountain, 1298 m, 23°03′54″N 104°50′20″E, 23 June 2020, *Chu Xuan Canh Chuong Duc Thanh, & Pham Van The, PVT1009* (VNMN!)

## The updated checklist of Ranunculaceae in Vietnam

Nearly 17 years since the last publication of *Nguyen* in 2003, this newest checklist records 11 genera, 45 species and two varieties of Ranunculaceae in Vietnam according to *APG4 classification system (2016)*. Of which, one variety of *Aconitum*, one species of *Actaea*, four species of *Anemone*, 18 species and one variety of *Clematis*, one species of *Consolida*, three species of *Coptis*, two species of *Delphinium*, two species of *Dichocarpum*, four species of

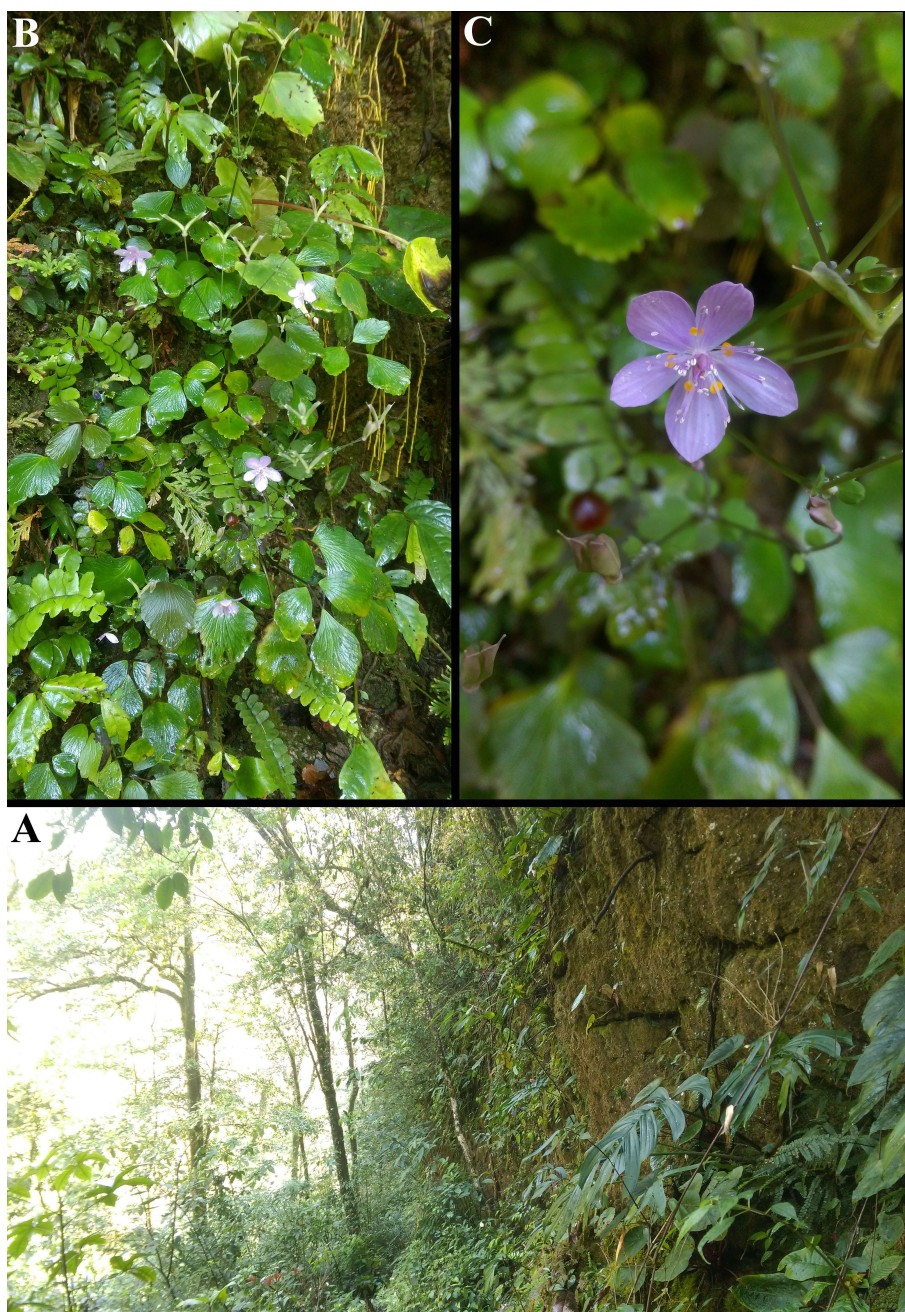

**Figure 4** *Dichocarpum hagiangense* ense L.K. Phan & V.T. Pham in its natural habitat. (A) Rocky cliff at the limestone forest. (B) Plant with inflorescence. (C) Flower.

*Naravelia*, eight species of *Ranunculus*, and two species of *Thalictrum*. Although four species *Naravelia dasyoneura* Korth., *N. laurifolia* Wall. ex Hook.f. & Thomson, *N. siamensis* Craib, and *Ranunculus blumei* Steud. are recorded in the checklist but their taxonomic revision is recommended.

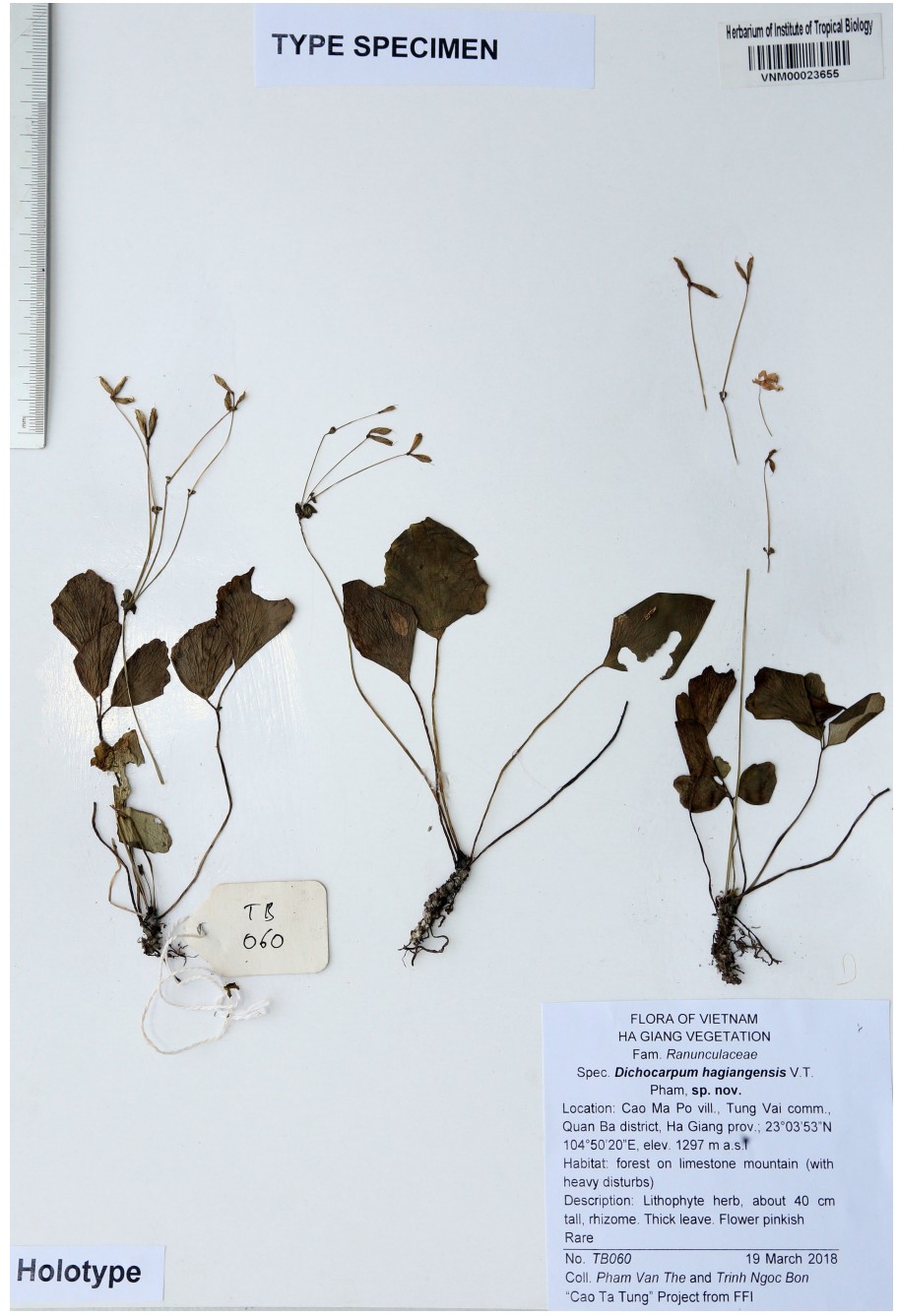

**Figure 5** Holotype of *Dichocarpum hagiangense* ense L.K. Phan & V.T. Pham at VNM herbarium.

Each species or variety in the checklist is provided with an accepted scientific name, followed by origin publication, and literature or other names and literature which were recorded for Vietnam.

*Aconitum carmichaelii* var. *truppelianum* (Ulbr.) W.T. Wang & P.K.Hsiao, *in* Fl. Reipubl. Popul. Sin. 27: 268 1979; Pham, An Illust. Fl. Vietnam 1: 325 1999. - *A. fortunei* Hemsl., J.

**Table 2 Comparison of diagnostic features of *Dichocarpum hagiangense* with *D. trifoliolatum*, *D. basilare*, and *D. carinatum*.**

| Characteristic | D. hagiangense | D. trifoliolatum | D. basilare | D. carinatum |
|---|---|---|---|---|
| **Rhizome** | | | | |
| Length (cm) | 4–9 | 16 | 1 | 8–10 |
| Diameter (cm) | 0.5–0.8 | 0.4 | 0.6 | 0.5–0.6 |
| **Leaf** | | | | |
| Number | 4–6 | 3 | 3–5 | 2 |
| Foliated | (2–)3 or simple | 3 or simple | (3–)5 | 12–15 |
| Petiole length (cm) | 3.3–10.5 | 6.2–8.3 | 2–4.7 | to 12 |
| **Basal leaf** | | | | |
| Central leaflet size (cm) | 3.0–4.0 × 2.4–2.8 | 3.7–4.3 × 2.3–2.8 | 1.2–2.7 × 0.8–2.8 | 1.8–2.8 × 0.9–2 |
| Central leaflet apex | retuse | rounded | obtuse | obtuse |
| Leaflet marin | distally crenate | distally crenate | distally crenate | 3-lobed |
| **Stem leaf present** | yes | no | no | no |
| **Inflorescence** | | | | |
| Type | dichasial | monochasial | – | – |
| Flowered | 2–6 | 3 | 3–5 | 3–5 |
| Flowering stem height (cm) | 9.5–14.5 | 23–25 | 6–19 | – |
| Pedicel length (cm) | 1.7–9.0 | 0.4–1.7 | – | – |
| **Sepal** | | | | |
| Color | pink purple | pinkish | white | pinkish |
| Length (mm) | 10.5–11.5 | 3.5 | – | – |
| **Petal limb** | | | | |
| Shape | broadly obcordate | flabellate | – | – |
| Length (mm) | 1.2–1.4 | 2.5 | – | – |
| Apex | obcordate | retuse | – | – |
| **Stamen** | | | | |
| Number | 30–40 | – | – | – |
| Length (mm) | 3–4 | 3.5 | – | – |
| **Follicle** | | | | |
| Length (mm) | 10–14 | 8–10 | 7.5–10 | – |
| Persistent styles length (mm) | 2 | 2.5 | 1.5 | – |
| **Seed** | | | | |
| Shape | globose | ellipsoid | subglobose | subglobose |
| Diam. (mm) | 0.7 | 2.5 | 1.5 | 1 |

Linn. Soc., Bot. 23: 20 1886; Gagnepain, Supplément a la Flore générale de l'Indo-Chine 1: 16 1938; Nguyen, Checkl. Pl. Spec. Vietnam 2: 154 2003.

*Actaea cordifolia* DC., Syst. Nat. 1: 383 1817. - *Cimicifuga racemosa* var. *cordifolia* (DC.) A.Gray, Syn. Fl. N. Amer. 1(1): 55 1895; Pham, An Illust. Fl. Vietnam 1: 324 1999.

*Anemone chapaensis* Gagnep., Bull. Soc. Bot. France 76: 315 1929; Gagnepain, Supplément a la Flore générale de l'Indo-Chine 1: 11 1938; Pham, An Illust. Fl. Vietnam 1: 320 1999; Nguyen, Checkl. Pl. Spec. Vietnam 2: 154 2003.

*Anemone poilanei* Gagnep., Bull. Soc. Bot. France 76: 315 1929; Gagnepain, Supplément a la Flore générale de l'Indo-Chine 1: 11 1938; Pham, An Illust. Fl. Vietnam 1: 321 1999; Nguyen, Checkl. Pl. Spec. Vietnam 2: 155 2003.

*Anemone rivularis* Buch.-Ham. ex DC., Syst. Nat. 1: 211 1817; Gagnepain, Supplément a la Flore générale de l'Indo-Chine 1: 9 1938; Pham, An Illust. Fl. Vietnam 1: 321 1999; Nguyen, Checkl. Pl. Spec. Vietnam 2: 155 2003.

*Anemone sumatrana* de Vriese, Pl. Jungh. 76 1851; Gagnepain, Supplément a la Flore générale de l'Indo-Chine 1: 9 1938; Pham, An Illust. Fl. Vietnam 1: 321 1999; Nguyen, Checkl. Pl. Spec. Vietnam 2: 155 2003.

*Clematis armandii* Franch. Nouv. Arch. Mus. Hist. Nat. II, 8: 184 1885; Finet & Gagnepain, Flore générale de l'Indo-Chine 1: 3 1907; Pham, An Illust. Fl. Vietnam 1: 315 1999; Nguyen, Checkl. Pl. Spec. Vietnam 2: 155 2003.

*Clematis brevicaudata* DC., Syst. Nat. 1: 138 1817; Gagnepain, Supplément a la Flore générale de l'Indo-Chine 1: 4 1938; Pham, An Illust. Fl. Vietnam 1: 318 1999; Nguyen, Checkl. Pl. Spec. Vietnam 2: 155 2003.

*Clematis buchananiana* DC., Syst. Nat. 1: 140 1817. - *Clematis bucamara* Buch.-Ham. ex DC., Syst. Nat. 1: 140 1817; Pham, An Illust. Fl. Vietnam 1: 316 1999; Nguyen, Checkl. Pl. Spec. Vietnam 2: 155 2003.

*Clematis cadmia* Buch.-Ham. ex Hook.f. & Thomson, Fl. Brit. India 1: 2 1872; Finet & Gagnepain, Flore générale de l'Indo-Chine 1: 7 1907; Gagnepain, Supplément a la Flore générale de l'Indo-Chine 1: 6 1938; Pham, An Illust. Fl. Vietnam 1: 316 1999; Nguyen, Checkl. Pl. Spec. Vietnam 2: 156 2003.

*Clematis chinensis* Osbeck, Dagb. Ostind. Resa 205 1757; Finet & Gagnepain, Flore générale de l'Indo-Chine 1: 5 1907; Gagnepain, Supplément a la Flore générale de l'Indo-Chine 1: 6 1938; Pham, An Illust. Fl. Vietnam 1: 316 1999; Nguyen, Checkl. Pl. Spec. Vietnam 2: 156 2003.

*Clematis fasciculiflora* Franch., Pl. Delavay. 5 1889; Gagnepain, Supplément a la Flore générale de l'Indo-Chine 1: 3 1938; Pham, An Illust. Fl. Vietnam 1: 316 1999; Nguyen, Checkl. Pl. Spec. Vietnam 2: 156 2003.

*Clematis florida* Thunb., Syst. Veg. ed. 14 512 1784. - *Anemone japonica* Houtt., Nat. Hist. 2(9): 191 1778; Gagnepain, Supplément a la Flore générale de l'Indo-Chine 1: 8 1938; Pham, An Illust. Fl. Vietnam 1: 321 1999; Nguyen, Checkl. Pl. Spec. Vietnam 2: 154 2003.

*Clematis fulvicoma* Rehder & E.H.Wilson, Pl. Wilson. 1: 327 1913; Pham, An Illust. Fl. Vietnam 1: 316 1999; Nguyen, Checkl. Pl. Spec. Vietnam 2: 156 2003.

*Clematis gialaiensis* Serov, Bot. Zhurn. (Moscow & Leningrad) 79(7): 106 1994; Pham, An Illust. Fl. Vietnam 1: 319 1999; Nguyen, Checkl. Pl. Spec. Vietnam 2: 156 2003.

*Clematis gouriana* Roxb. ex DC., Syst. Nat. 1: 138 1817; Pham, An Illust. Fl. Vietnam 1: 319 1999. - *Clematis vitalba* var. *gouriana* (Roxb. ex DC.) Finet & Gagnep., Bull. Soc. Bot. France 50: 532 1903; Pham, An Illust. Fl. Vietnam 1: 319 1999.

*Clematis hagiangense* N.T. Do, Acta Phytotax. Sin. 44: 595 2006.

*Clematis henryi* Oliv., Hooker's Icon. Pl. 19: t. 1819 1889; Gagnepain, Supplément a la Flore générale de l'Indo-Chine 1: 6 1938; Pham, An Illust. Fl. Vietnam 1: 316 1999; Nguyen, Checkl. Pl. Spec. Vietnam 2: 157 2003.

*Clematis leschenaultiana* DC., Syst. Nat. 1: 151 1817; Finet & Gagnepain, Flore générale de l'Indo-Chine 1: 6 1907; Pham, An Illust. Fl. Vietnam 1: 315 1999; Nguyen, Checkl. Pl. Spec. Vietnam 2: 157 2003.

*Clematis loureiroana* DC. Syst. Nat. 1: 144 1817; Pham, An Illust. Fl. Vietnam 1: 316 1999; Nguyen, Checkl. Pl. Spec. Vietnam 2: 157 2003.

*Clematis meyeniana* var. *granulata* Finet & Gagnep., Bull. Soc. Bot. France 50: 530 1903; Finet & Gagnepain, Flore générale de l'Indo-Chine 1: 4 1907; Gagnepain, Supplément a la Flore générale de l'Indo-Chine 1: 3 1938; Pham, An Illust. Fl. Vietnam 1: 316 1999. - *Clematis granulata* (Finet & Gagnep.) Ohwi; Acta Phytotax. Geobot. 6: 147 1937; Pham, An Illust. Fl. Vietnam 1: 316 1999; Nguyen, Checkl. Pl. Spec. Vietnam 2: 156 2003.

*Clematis smilacifolia* Wall., Asiat. Res. 13: 402 1820; Finet & Gagnepain, Flore générale de l'Indo-Chine 1: 3 1907; Gagnepain, Supplément a la Flore générale de l'Indo-Chine 1: 3 1938; Pham, An Illust. Fl. Vietnam 1: 318 1999; Nguyen, Checkl. Pl. Spec. Vietnam 2: 157 2003. - *Clematis petelotii* Gagnep., Notul. Syst. (Paris) 15: 36 1955; Pham, An Illust. Fl. Vietnam 1: 318 1999; Nguyen, Checkl. Pl. Spec. Vietnam 2: 157 2003. - *Clematis subpeltata* Wall., Pl. Asiat. Rar. 1: 19 1829; Pham, An Illust. Fl. Vietnam 1: 318 1999.

*Clematis subumbellata* Kurz, J. Asiat. Soc. Bengal, Pt. 2, Nat. Hist. 39(2): 61 1870. - *Clematis umbellifera* Gagnep., Bull. Soc. Bot. France 82: 477 1935 publ. 1936; Gagnepain, Supplément a la Flore générale de l'Indo-Chine 1: 4 1938; Pham, An Illust. Fl. Vietnam 1: 319 1999

*Clematis uncinata* Champ. ex Benth., Hooker's J. Bot. Kew Gard. Misc. 3: 255 1851; Finet & Gagnepain, Flore générale de l'Indo-Chine 1: 2 1907; Gagnepain, Supplément a la Flore générale de l'Indo-Chine 1: 3 1938; Pham, An Illust. Fl. Vietnam 1: 319 1999; Nguyen, Checkl. Pl. Spec. Vietnam 2: 157 2003.

*Clematis vietnamensis* W.T.Wang & N.T.Do, Acta Phytotax. Sin. 44: 680 2006.

*Consolida ajacis* (L.) Schur, Verh. Mitth. Siebenbürg. Vereins Naturwiss. Hermannstadt 4(3): 47 1853. - *Delphinium ajacis* L., Sp. Pl. 531 1753; Pham, An Illust. Fl. Vietnam 1: 324 1999; Nguyen, Checkl. Pl. Spec. Vietnam 2: 159 2003.

*Coptis chinensis* Franch., J. Bot. (Morot) 11: 231 1897; Pham, An Illust. Fl. Vietnam 1: 325 1999; Nguyen, Checkl. Pl. Spec. Vietnam 2: 158 2003.

*Coptis quinquesecta* W.T.Wang, Acta Phytotax. Sin. 6: 219 1957; Nguyen, Checkl. Pl. Spec. Vietnam 2: 158 2003.

*Coptis teeta* Wall., Trans. Med. Soc. Calcutta 8: 87 1836; Pham, An Illust. Fl. Vietnam 1: 325 1999; Nguyen, Checkl. Pl. Spec. Vietnam 2: 158 2003.

*Delphinium ambiguum* L., Sp. Pl. ed. 2 749 1762. - *Delphinium nanum* DC., Syst. Nat. 1: 349 1817; Pham, An Illust. Fl. Vietnam 1: 325 1999.

*Delphinium anthriscifolium* Hance, J. Bot. 6: 207 1868; Gagnepain, Supplément a la Flore générale de l'Indo-Chine 1: 14 1938; Pham, An Illust. Fl. Vietnam 1: 324 1999; Nguyen, Checkl. Pl. Spec. Vietnam 2: 157 2003.

*Dichocarpum dalzielii* (J.R.Drumm. & Hutch.) W.T.Wang & P.K.Hsiao, Acta Phytotax. Sin. 9: 327 1964; *Phan, Averyanov & Nguyen, 2001*.

*Dichocarpum sutchuenense* (Franch.) W.T.Wang & P.K.Hsiao, Acta Phytotax. Sin. 9: 328 1964. Pham, An Illust. Fl. Vietnam 1: 324 1999.

*Naravelia dasyoneura* Korth., Ned. Kruidk. Arch. 1: 208 1848; Finet & Gagnepain, Flore générale de l'Indo-Chine 1: 8 1907; Gagnepain, Supplément a la Flore générale de l'Indo-Chine 1: 7 1938; Pham, An Illust. Fl. Vietnam 1: 319 1999; Nguyen, Checkl. Pl. Spec. Vietnam 2: 159 2003.

*Naravelia laurifolia* Wall. ex Hook.f. & Thomson, Fl. Ind. 1: 3 1855; Pham, An Illust. Fl. Vietnam 1: 320 1999; Nguyen, Checkl. Pl. Spec. Vietnam 2: 159 2003.

*Naravelia siamensis* Craib, Bull. Misc. Inform. Kew 1915: 419 1915; Pham, An Illust. Fl. Vietnam 1: 320 1999; Nguyen, Checkl. Pl. Spec. Vietnam 2: 160 2003.

*Naravelia zeylanica* (L.) DC. Syst. Nat. 1: 167 1818; Finet & Gagnepain, Flore générale de l'Indo-Chine 1: 8 1907; Gagnepain, Supplément a la Flore générale de l'Indo-Chine 1: 7 1938; Pham, An Illust. Fl. Vietnam 1: 320 1999; Nguyen, Checkl. Pl. Spec. Vietnam 2: 160 2003. - *Atragene zeylanica* L., Sp. Pl. 542 1753; Pham, An Illust. Fl. Vietnam 1: 320 1999.

*Ranunculus blumei* Steud., Nomencl. Bot. ed. 2, 2: 432 1841; Pham, An Illust. Fl. Vietnam 1: 322 1999.

*Ranunculus cantoniensis* DC., Prodr. 1: 43 1824; Pham, An Illust. Fl. Vietnam 1: 322 1999; Nguyen, Checkl. Pl. Spec. Vietnam 2: 160 2003.

*Ranunculus diffusus* DC., Prodr. 1: 38 1824; Gagnepain, Supplément a la Flore générale de l'Indo-Chine 1: 12 1938; Pham, An Illust. Fl. Vietnam 1: 323 1999; Nguyen, Checkl. Pl. Spec. Vietnam 2: 160 2003.

*Ranunculus japonicus* Langsd. ex DC., Prodr. [A. P. de Candolle] 1: 38 1824; Finet & Gagnepain, Flore générale de l'Indo-Chine 1: 10 1907; Gagnepain, Supplément a la Flore générale de l'Indo-Chine 1: 12 1938.

*Ranunculus pensylvanicus* L. f., Suppl. Pl. 272 1781; Finet & Gagnepain, Flore générale de l'Indo-Chine 1: 10 1907; Gagnepain, Supplément a la Flore générale de l'Indo-Chine 1: 13 1938; Pham, An Illust. Fl. Vietnam 1: 323 1999; Nguyen, Checkl. Pl. Spec. Vietnam 2: 160 2003.

*Ranunculus sceleratus* L., Sp. Pl. 551 1753.; Finet & Gagnepain, Flore générale de l'Indo-Chine 1: 11 1907; Gagnepain, Supplément a la Flore générale de l'Indo-Chine 1: 13 1938; Pham, An Illust. Fl. Vietnam 1: 323 1999; Nguyen, Checkl. Pl. Spec. Vietnam 2: 161 2003.

*Ranunculus silerifolius* H. Lév., Repert. Spec. Nov. Regni Veg. 7(146–148): 257 1909; Pham, An Illust. Fl. Vietnam 1: 323 1999; Nguyen, Checkl. Pl. Spec. Vietnam 2: 161 2003.

*Ranunculus sundaicus* (Backer) H.Eichler, Biblioth. Bot. 124: 94 1958; Pham, An Illust. Fl. Vietnam 1: 323 1999.

*Thalictrum foliolosum* DC., Syst. Nat. 1: 175 1817; Pham, An Illust. Fl. Vietnam 1: 322 1999; Nguyen, Checkl. Pl. Spec. Vietnam 2: 161 2003.

*Thalictrum ichangense* Lecoy. ex Oliv., Hooker's Icon. Pl. 18(3): pl. 1765 1888; Gagnepain, Supplément a la Flore générale de l'Indo-Chine 1: 7 1938; Pham, An Illust. Fl. Vietnam 1: 322 1999; Nguyen, Checkl. Pl. Spec. Vietnam 2: 161 2003.

## DISCUSSION

According to a recent report, *Dichocarpum* species usually have potential value for pharmacy (*Hao, 2018*), therefore this research could open a chance for medicinal herb

studying. Besides, the new species was found in the forest of limestone mountain where some new and interesting plant species were recorded in recent years such as *Paraboea villosa* (Gesneriaceae), *Loropetalum flavum* (Hamamelidaceae), Magnolias, or Orchids (*Tu, Nguyen & Nguyen, 2015*; *Averyanov et al., 2018*; *Averyanov et al., 2019*; *Averyanov et al., 2020*). Also, a Vietnam's second-largest population of Critically Endangered Tonkin Snub-nosed Monkey (*Rhinopithecus avunculus*) with about 15–21 individuals has been recorded in this area (*Vietnam Academicy of Science and Technology, 2007*; *Le, 2010*; *Schwitzer et al., 2015*; *Nguyen, Pham & Le, 2016*; *Quyet et al., 2020*). Despite the high value of biodiversity, the natural forest is strongly impacted by large-scale deforestation for the cultivation of Tsao-ko Cardamom (*Amomum tsao-ko*) and Ling Xiang Cao (*Lysimachia foenum-graecum*) (*Le, 2010*). For this reason, this study could give additional scientific value for the provincial manager's decision for planning protection of this forest as the establishment of a "Species and Habitat Conservation Area" and application of community-based forest conservation for long-term sustainable biodiversity conservation.

## CONCLUSION

With this discovery, a total of ca. 20 species of the genus Dichocarpum are known, and three species are recorded for Vietnam. On the other hand, the checklist of the Ranunculaceae of Vietnam is a good reference for oversea researchers while limited international language literature from the country. In contrast, the species from Lao Cai province with label no. HAL 2212 (HN, LE, MO) is needed to recollect to determine the exact species name.

## ACKNOWLEDGEMENTS

The authors cordially thank Dr. Andrey Erst and an anonymous reviewer for their helpful comments. We would like to express our thanks to MSc. Nguyen Van Truong, Mr. Dao Cong Anh, Mrs. Dinh Thi Kim Van, Mr. Chuong Duc Thanh and Mr. Chu Xuan Canh from Fauna & Flora International—Vietnam Programme for their field survey assistance and for arranging the fieldwork. Virtual Herbaria LE, MO and P are also highly acknowledged.

### Funding

This study was supported by Fauna and Flora International - Vietnam Programme; and Center for Resources, Environment and Climate change on the "Investigation and Assessment on the Flora diversity and Vegetation in Cao Ma Po–Ta Van–Tung Vai forest, Quan Ba district, Ha Giang province" (March to April 2018) . The funders had no role in study design, data collection and analysis, decision to publish, or preparation of the manuscript.

### Grant Disclosures

The following grant information was disclosed by the authors:
Fauna and Flora International - Vietnam Programme.

Center for Resources, Environment and Climate change on the "Investigation and Assessment on the Flora diversity and Vegetation in Cao Ma Po–Ta Van–Tung Vai forest Quan Ba district, Ha Giang province.

## Competing Interests

The authors declare there are no competing interests.

## Author Contributions

- Minh Ty Nguyen analyzed the data, prepared figures and/or tables, authored or reviewed drafts of the paper, and approved the final draft.
- Ngoc Bon Trinh, Thanh Thi Viet Tran and Tran Duc Thanh analyzed the data, prepared figures and/or tables, and approved the final draft.
- Long Ke Phan and Van The Pham conceived and designed the experiments, performed the experiments, analyzed the data, prepared figures and/or tables, authored or reviewed drafts of the paper, and approved the final draft.

## Field Study Permissions

The following information was supplied relating to field study approvals (i.e., approving body and any reference numbers):

Collection permits were issued by the "Forest Protection Department of Ha Giang province" (applied by Fauna & Flora International - Vietnam Programme).

## DNA Deposition

The following information was supplied regarding the deposition of DNA sequences:

The group *Dichocarpum* and *Isopyrum manshuricum* sequences are available at GenBank: KY235682, KY235683, KY235690, KY235691, KY235684, HQ844055, HQ844062, KY235688, KY235685, EF437116, HQ727692, KY235686, EF437115, KY235694, KY235695, KY235696, KY235697, KY235698, KY235692, KY235693, HQ844053, HQ844063, KY235690, KY235691, MT739412, EF437119.

## Data Availability

Data is available at FigShare: Pham, Van The (2020): Raw data for description and measurement of the new species. figshare. Figure. https://doi.org/10.6084/m9.figshare.11844915.v1.

The specimens are deposited at the Herbarium of Institute of Tropical Biology (VNM) with code VNM00023655, and Herbarium of Komarov Botanical Institute (LE) with code LE01049587.

## New Species Registration

The following information was supplied regarding the registration of a newly described species:

*Dichocarpum hagiangense* L.K. Phan & V.T. Pham: 77211047-1.

## Supplemental Information

Supplemental information for this article can be found online at http://dx.doi.org/10.7717/peerj.9874#supplemental-information.

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
