# Peer review of "Dichocarpum hagiangense—a new species and an updated checklist of Ranunculaceae in Vietnam"

_PeerJ, doi:10.7717/peerj.9874_

## Round 0.1 · original submission · Major Revisions

Dear Dr. Pham,

I have received 2 reviews on your ms. Both recommend major revisions. As a major point both ask to include DNA data of the new taxon. I agree that it would indeed much strengthen the study. Therefore, it makes sense to do this. There several other points in special made by reviewer 2 which should be relatively easy to implement.

Overall, major revisions are necessary.

Best wishes
Mike Thiv

Reviewer 1 ·

Basic reporting

Comments to the Author
This manuscript is dedicated to describe a new species, Dichocarpum hagiangensis, and to update the checklist of Ranunculaceae in Vietnam.
When reviewing the manuscript, several questions and suggestions arose, so here they are.
1. The authors only provide morphological evidence for the new species. Based on Table 1, most of morphological characters are quantitative and are overlapped among those three species. I strongly recommend adding molecular evidence. Xiang et al. (2017) have presented a phylogenetic analysis for Dichocarpum based on four DNA regions. It is therefore very easy to do this. The same four markers for this new species can be sequenced, at least ITS, trnH-psbA, and matK should be generated since they are usually used as DNA barcoding sequences. I believe molecular data can provide more robust support for your new species if it is indeed a good species.
2. L65: change “and” to “or”).
3. L145: change “ca” to “ca.”.
4. L204: change “follows” to “followed”(language, “by”).
5. The reference should be arranged alphabetically, please notice “Liesner R. 1995.”
6. L406: change “-” to “–”.

Experimental design

'no comment'

Validity of the findings

'no comment'

·

Basic reporting

See the main comments in the text of the review.
In general, everything is fine, but serious changes are required!

Experimental design

See the main comments in the text of the review.
In general, everything is fine, but serious changes are required!

Validity of the findings

See the main comments in the text of the review.
In general, everything is fine, but serious changes are required!

Additional comments

Reviewing Manuscript 45699v1

Dichocarpum hagiangensis - a new species and an updated checklist of Ranunculaceae in Vietnam

The research of the biodiversity and revealing the new species is the actual and significant problem, and the investigations, aimed on resolving of this problem, are to be warmly welcomed.
The paper is a classical description of a new taxon based on morphological characters. The new species is gorgeous and does seem to be different from the closely related ones mentioned in the manuscript.

The paper could be more interesting and notice by more readers if the more general information in the introduction and conclusion be given, e.g.: newest work about Dichocarpum, number of species in the genus in World, Asia, another regions, a key or at least a checklist of all the all species (with new morphological characters), distribution map of a new species and all species that are relatives of a new species (!) or distribution map new species and all species Dichocarpum from Vietnam. It will be much more informative and useful for many other scientists.
These are general comments!

The main comments which need to be corrected by the authors include:

1. Diagnosis should be changed:
According morphological characters new species belong to Sect. Dichocarpum, subsect. Dalzielia. Dichocarpum hagiangensis is morphologically most similar to D. trifoliata, but differs in flowers diameter (2-2.3 vs 0.7 cm), longer sepals (10.5-11.5 vs 3.5 mm), number, tooth shape of basal leaves and rhizome diameter (more 0.5 vs 0.4 cm) and dichasial inflorence. However, D. hagiangensis differs from D. basilare in having the 3-5-foliolate leaves and retuse central leaflet apex. And compare with D. carinatum please (because in Xiang et al. 2016 all related species have been included to the one clade)!!!!!

2. Change the terms in table and in text. All leaves is compound. Change “leaf” to “basal leaf” and delete “compound leaves”, change “flower number” to flowers number and add stem leaves parameters. Because new species have stem leaves and bract too. It is also necessary to add the characteristics of stem leaves in the description of the taxon.

3. Make a distribution map of all species of the genus in Vietnam! Use your data, literature data and herbarium funds.
For example (Material and methods): Maps of records will be made with SimpleMappr (http://www.simplemappr.net).

It is necessary to do all the calculations and add information to the sections materials and methods and results.

4. IUCN conservation assessments can only be made with the evidence available at the time of study so it is ingenuous to say that more information may lead to a better assessment in future. It is possible to calculate the EOO and AOO, and to make a full assessment. This should be done. Naturally, more information may come to hand in future but there’s no need to wait before making an assessment.

For example: Conservation analysis was performed using criteria from the International Union for the Conservation of Nature (IUCN 2019). The Extent of Occurrence (EOO) and Area of Occupancy (AOO) of each species were estimated using GeoCat (Bachman et al. 2011).

Literature: Bachman S, Moat J, Hill A, de la Torre J, Scott B (2011) Supporting Red List threat assessments with GeoCAT: Geospatial conservation assessment tool. ZooKeys 150: 117–126. https://doi.org/10.3897/zookeys.150.2109

It is necessary to do all the calculations and add information to the sections materials and methods and results.

5. It is necessary to add a scanned sample of a herbarium of a standard sample with a barcode to the text of the article.

6. If you have not done a DNA analysis of your samples and have not compared with other species, then you need to discuss the position of the species relative to other species. First of all, it will find out the kinship by morphology, then on the basis of known data on DNA analysis, it is necessary to conduct a discussion on the differences and similarities with other types of close kinship. It is imperative to make references to molecular studies and to phylogenetic trees.

In spite of this major changes are needed! Make all changes to the text, according to my comments.
Finally, I would like to ask you to pay close attention to the formatting guidelines of PeerJ. I have noticed that several parts of the text do not meet those requirements. After analysing MS, I am pleased to inform that my decision is to accept your submission, but with the need of Major Revision.

---

## Round 0.2 · Major Revisions

Dear Dr. Pham,

I - and the reviewers - asked you to include DNA data of the new taxon because it would much strengthen the study.

I understand that corona virus makes everything more difficult, but you can easily use a very small part of the type herbarium specimen(s) without going to the field. It is freshly collected in 2018 and should likely work for PCR. So I think you should give this a try.

Best wishes
Mike Thiv

---

## Round 0.3 · accepted · Accept

Dear authors,

I received a positve feedback on your revised version from a reviewer.
I therefore accept your ms. after some minor editorial changes.
I indicated them in the attached pdf, they are marked in lilac.

Best wishes
Mike Thiv

·

Basic reporting

All corrections were made according to the recommendations of the reviewers, the manuscript was significantly revised. I recommend to accept.

Experimental design

ok

Validity of the findings

ok